# Requirements of Modern Russian Agricultural Production for Digital Competencies of an Agricultural Specialist

**Elena Khudyakova** [1]**, Alexandra Shitikova** [2] **, Marina Nikolaevna Stepantsevich** [2] **and Anastasia Grecheneva** [1,*]

1   Department of Applied Mathematics, Russian State Agrarian University—Moscow Timiryazev Agricultural Academy, Timiryazevskaya Street, 49, 127550 Moscow, Russia
2   Department of Plant Growing and Meadow Systems, Russian State Agrarian University—Moscow Timiryazev Agricultural Academy, Timiryazevskaya Street, 49, 127550 Moscow, Russia
*   Correspondence: a.grechenevaav@rgau-msha.ru; Tel.: +7-(919)019-36-80

**Abstract:** The purpose of the study is to determine the needs of modern Russian agriculture for specialists with certain, most-in-demand, digital competencies. The research methodology is based on the application of the expert assessments method, the method of random statistical selection of experts, and the scientific generalization method. The field of the research is modern digital technologies in agriculture, as well as the corresponding competencies of Russian agricultural university graduates. The study period is 2021–2022. Having acquired competencies of a modern agricultural specialist at the university should help graduates to integrate into the production process as quickly as possible, the latter undergoing qualitative changes due to the transition to a new technological order based on the use of digital technologies. The study of the current curricula of Russian agricultural universities has shown their lack of adequacy regarding the modern requirements of agricultural production. It indicates the need to examine the curricula. Taking into account the fact that digital technologies are rapidly improving and being introduced into agricultural production, further research in this area should be conducted annually in order to increase the compliance of training at universities according to modern requirements of agricultural production.

**Keywords:** digital technologies in Russian agriculture; educational programs of Russian universities; competencies of agricultural specialists

## 1. Introduction

Digitalization is one of the strategic directions for the development of the modern Russian economy in general and in agriculture in particular. It is to ensure economic growth in the industry, and promote sustainable development of agriculture. In accordance with the national program "Digital Economy of the Russian Federation" [1] and the departmental project "Digital Agriculture" [2], one of the key factors in the implementation of the agro-industrial complex digitalization programs, along with such blocks as digital technologies, regulatory regulation, digital infrastructure, and information security, is appropriate staffing. According to the national program "Digital Economy of the Russian Federation", its implementation requires about 120 thousand university graduates skilled in information and telecommunication technologies, and the share of the population with digital skills should be about 40% (they should be able to work with information obtained from the internet, have skills in working in the internet system, use various internet services, work with application software products and others).

In recent decade, Russian agriculture has been demonstrating a steady growth in the production of its main products, according to Spearman's test. This is largely due to the growth in labor productivity and the use of precision farming technologies that reduce unit production costs and increase crop yields. For this reason, the educational programs of universities, in particular in agronomy, should be updated with reference to new digital

technologies at a minimum of once a year. A survey of employees working for leading digital agricultural companies and top managers of the main Russian agricultural holdings have shown that educational programs must be advanced towards the development of the following competencies, according to which graduates of agricultural universities should:

- Know the main trends in the development of modern digital technologies in crop production;
- Use information systems to improve production plans, control operation processes, as well as to prepare reports;
- Know the best practices in the application of both national and international digital technologies in agronomy;
- Possess skills in working with information systems for managing agricultural production (ERP-systems);
- Use mobile applications for visual crop control;
- Analyze weather forecast data from detectors, sensors and other smart devices, etc.

The recent trend in the development of Russian agriculture can be characterized as a steady growth.

A comparison of the development stability of agriculture in the Russian Federation in 2021 with 2010's indicators using Spearman's test (ρ) shows a steady growth in the main products in agricultural organizations (except for grain) (Table 1).

**Table 1.** Sustainability of production of the main types of agricultural products in the Russian Federation (2010–2021).

| Food * | Spearman's Test [1] | Growth Sustainability |
|---|---|---|
| Grain | −0.74 | unsteady decline |
| Sugar beet | 0.94 | steady growth |
| Eggs | 1.0 | steady growth |
| Meat and poultry | 1.0 | steady growth |
| Milk | 1.0 | steady growth |

[1] critical test value—0.85. *—These products have been chosen because they are not the main ones in the human diet [3].

However, in relation to the levels of 1990, most types of products have shown a steady decline in production.

Digital technologies are designed to increase the sustainability of agricultural production. The implementation of digital technologies is characterized by the following effects (Table 2).

**Table 2.** Effects of digital technologies on agricultural production [4].

| Direct Effect | Indirect Effect |
|---|---|
| - Cost savings due to the replacement of manual labor and accounting for consumed raw materials;<br>- Increase in volumes, faster processing of information and acceptance of administrative decisions;<br>- Saving of labor hours, etc. | - Improvement of enterprise performance indicators;<br>- Increase in the number of clients (contractors);<br>- Gain in sales or market segment;<br>- Faster paperwork and provision of services, etc. |

The scientific literature widely discusses methodological, economic, technical and philosophical issues of the global economy digitalization [5–9].

However, the lack of a sufficient number of specialists with digital competencies is one of the critical obstacles to the introduction of digital technologies. Each field of activity in a certain period of productive forces development, in particular, digital technologies, needs certain kinds of competencies (in this case, digital ones). The objective of the research

was to study the structure of the forecasted need for agronomic specialists with professional competencies in the application of digital technologies in agronomy.

In particular, there is a shortage of qualified agronomic personnel in the country. There was an objective need to introduce new specialties into the high school educational process, providing training for specialists in the implementation of agricultural production and the use of end-to-end digital technologies (big data, neuro-technologies, augmented reality technologies, etc.) The current realities of the global and Russian agricultural sector pose one of the key tasks for universities—the need to develop and update existing educational programs using new indicators for achieving digital competencies. The issue of directions and methods for the formation of digital competencies among university graduates in connection with the digital transformation of the economies of most countries of the world is relevant and widely discussed in various scientific publications [10–13].

At present, it is very important to organize cooperation between universities and organizations that develop digital technologies so that university graduates will be able to work under new conditions [14,15].

Therefore, it is essential to propose new forms of network interaction between universities and enterprises of the real sector of the economy [16,17].

Current curricula for agronomist training are based on professional standards that include digital competencies, but to a small extent. For example, in the professional standards of "agronomist", only a generalized formulation of digital technological abilities in agriculture can be found. It says that an agronomist should be able to "use specialized electronic information resources while collecting the data necessary for operational planning of work in crop production" and "apply geoinformation systems for operational planning of work in crop production." According to the competence "organization of the work of plant-growing teams in accordance with the technological maps of agricultural crops cultivation" in terms of knowledge, they should know "the composition, functions and possibilities of using information and telecommunication technologies in professional activities when organizing the work of plant-growing teams". According to the competency "control of the process of plant development during the growing season", graduates should be able to "use specialized electronic information resources and geoinformation systems when planning and monitoring plant development" [18]. However, this is not enough to be included in the training curricula of agronomists. Particular skills specification is needed and skills of using digital technologies that are in high demand today in Russian agriculture are to be highlighted. The transformation of the educational process in agricultural universities in accordance with the modern realities of agricultural production requires identification and classification of these new requirements based on a scientific approach, substantiating the graduate's competency model.

## 2. Materials and Methods

The following research methods were applied to achieve the goal—identification of the need for agronomists with professional competencies in the application of digital technologies in agronomy. The sustainability of the development of the production of basic food products was determined by the Spearman test, which was calculated by the formula:

$$\rho = 1 - \frac{5 \sum_{\tau=1}^{n} (R_i - S_i)^2}{n(n-1)(n+1)}, \tag{1}$$

where $R_i$ is the rank of the observation $x_i$ in the $x$ series; $S_i$ is the rank of observation $y_i$ in the $y$ series.

The relationship and correspondence between the educational programs of "agronomy" at national agricultural universities and the real need of the market and employers' requirements was analyzed by a scientific comparison method. (Comparison is a scientific method of cognition, in the process of its unknown (studied) phenomenon, objects are compared with already known, previously studied, in order to determine common features or differences between them).

The list of potential employers, including legal entities and/or individual entrepreneurs registered and operating in the Russian Federation for at least 5 years and having at least 100 full-time employees (since digital technologies are introduced mainly by large enterprises), was determined on the basis of random non-repetitive selection with a preliminary selection of homogeneous groups. The list of potential employers included companies: LLC "PhosAgro", agroholding "Dolgov Group", agro-industrial holding "MIRATORG", and LLC "KVS RUS".

To substantiate the conclusions, the results of an individual selective survey with pre-designed questions were used, the logic of which was consistent with the goal: "to analyze the needs of potential employers for specialists with professional competencies in digital crop production technologies." It was considered as a basis for building the competence model of the graduate.

The used methods of expert assessments and forecasting made it possible to predict the need for digital competencies in agricultural specialists for the next 5 years.

## 3. Results

The surveyed sample included 252 employees from 16 companies and enterprises of various organizational and legal forms, medium and large in size, with more than 100 employees. To determine the competitiveness of a graduate, the respondents were asked to evaluate the significance of some indicators from universities that train agronomists (Table 3).

**Table 3.** Competitiveness of graduates in agronomy (Bachelor's degree).

| No. | Indicators | Test Significance | | |
| --- | --- | --- | --- | --- |
| | | Highly Significant | Significant | Insignificant |
| 1 | The quality of training in agronomy | 84 | 16 | - |
| 2 | Share of employed graduates | 64 | 32 | 4 |
| 3 | Qualification of the teaching staff | 72 | 28 | - |
| 4 | Interaction with agricultural enterprises | 78 | 11 | 11 |
| 5 | Availability of modern facilities and resources | 62 | 38 | - |
| 6 | Use of digital technologies in education | 94 | 6 | - |

Representatives of the real sector of the economy consider such indicators as the use of digital technologies in education (94%) and the quality of student training (84%) as the most significant for the competitiveness of graduates.

On the other hand, employers think that interaction with agricultural enterprises (78%) and the qualification of teaching staff (72%) are noteworthy for competitiveness.

To analyze the needs of potential employers for professional competencies in the application of digital technologies in agronomy to update the bachelor's program in "agronomy", and "agribusiness", the authors compiled a list of potential employers (which may include legal entities and/or individual entrepreneurs registered and operating in the Russian Federation for at least 5 years and having at least 100 full-time employees) including the representativeness of the sample. The sample included the following companies: PhosAgro LLC, DolgovGroup Agricultural Holding, MIRATORG Agro-Industrial Holding, and KVS RUS LLC.

It was proposed to assess the degree of significance of a particular competence on a scale from 1 to 10, where 10 is the maximum demand for this competence, and 1 is the lowest demand for competence. The questions were open-ended—the specialists of the enterprises could indicate the competencies not included in the list proposed by them.

Further, potential employers were asked to evaluate the importance of several core digital competencies. Table 4 shows the findings of the survey.

**Table 4.** Significance of the digital competencies of an agronomist, according to the respondents.

| Competency | Mean Score (1 to 10) |
|---|---|
| Know the main trends in the development of modern digital technologies in crop production. | 8.93 |
| Use information systems to develop production plans, control work, and prepare reports. | 8.93 |
| Know the best practices of applying digital technologies in agronomy (in Russia and abroad). | 8.87 |
| Be skilled in information management systems for agricultural production (ERP-systems). | 8.73 |
| Use mobile applications for visual crop control. | 8.73 |
| Analyze weather forecast from sensors and other smart devices, as well as open access data to develop techniques to reduce the effect of limiting factors in the field crop productivity. | 8.53 |
| Know how to evaluate the effectiveness of the digital transformation of crop production. | 8.47 |
| Be skilled in compiling and analyzing electronic maps of differentiated application of fertilizers and plant protection products. | 8.40 |
| Develop methods for assessing and planning crop yields based on multivariate analysis. | 8.40 |
| Know the main features of end-to-end digital technologies and the possibilities of their application in crop production to develop a precision farming system. | 8.27 |
| Use information systems with the results of satellite imagery (or UAV) to create electronic maps of actual field contours. | 8.27 |
| Know the classification and main characteristics of vegetation indices (NDVI, SAVI, ARVI, EVI, LAI, NDWI, GNDVI, etc.); be able to use their data to develop elements of field crop cultivation technology and plant vegetation management. | 8.27 |
| Know characteristics and functionality of the main digital services, including geoinformation, to justify management decisions in crop production. | 8.20 |
| Be skilled in compiling and using electronic yield maps. | 8.20 |
| Know how to work with portable devices (N-Tester, Green Seeker, SPAD Chlorophyll Meter, and others) to make operational management decisions in crop production. | 8.07 |

A survey of potential employers identified the most sought-after competencies of graduates. The most significant competencies, according to employers (score from 8.5 to 8.93), were as follows:

- To know the main trends in the development of modern digital technologies in crop production;
- To use information systems to develop production plans, control work, and prepare reports;
- To recognize the best practices of applying digital technologies in agronomy (in Russia and abroad);
- To be skilled in information management systems for agricultural production (ERP-systems);
- To utilize mobile applications for visual crop control;
- To analyze weather forecast from sensors and other smart devices, as well as open access data to develop techniques to reduce the effect of limiting factors in the productivity of field crops;

- To be acquainted with effectiveness evaluation of the digital crop production transformation.

The competencies in Table 2 characterize digital hard skills and main digital professional skills of an agronomist. During the survey, respondents were also asked to evaluate what soft skills, transprofessional skills, are most in demand in the context of the digital transformation of the industry. The survey results are shown in Figure 1.

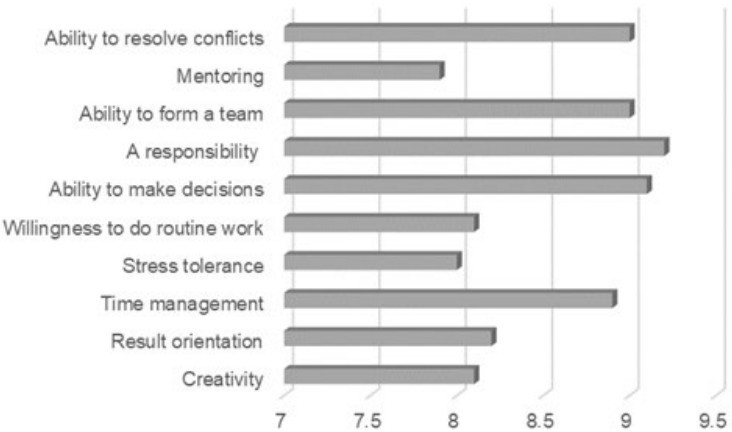

**Figure 1.** Scoring of digital competencies of a modern agricultural specialist (the assigned score is located along the abscissa axis).

This section may be divided by subheadings. It provides a concise and precise description of the experimental results, their interpretation, as well as the experimental conclusions that may be drawn.

Employers mostly look for such skills as time management, decision making, responsibility, team building, and conflict resolution.

The development of the competency model of the graduate also considered the recommendations of the industrial working group "Digital Technologies in Agriculture" of the Innopolis reference educational center, namely the basis for training a new generation of graduates in agronomy is the formation of learning outcomes in: analysis of information and process approach in agriculture; use of specialized information services for planning business processes; digital weather forecast stations and use of agrometeorological information in planning and predicting agronomic processes; geoinformation technologies for farming, observing crop rotations, and monitoring crops development; main types and varieties of soils using earth remote sensing (ERS) technologies; use of agricultural machines with modern digital navigation equipment for technological operations; dosing of agrochemicals based on remote sensing data; diagnostics, detection, and prediction of diseases and lack of nutrients for agricultural plants, and the identification of weeds and pests by using a database; landscape crop placement based on GIS technologies; field work reports based on special software and mobile applications; yield mapping systems; monitoring and accumulation of crop development data (field sensors, drone, webcam, navigator, smart weather station, etc.) using internet of things technologies. All these skills are of approximately equal importance for the quality of training of an agronomist with digital competencies.

The pilot project of the Ministry of Agriculture of the Russian Federation to train highly skilled agronomists of the new generation will be the implementation of an educational program based on the end-to-end agro-industrial digital technologies at the Russian State Agrarian University—Moscow Timiryazev Agricultural Academy, that will link all the educational courses of the curriculum and train bachelors and masters to solve the tasks set above (the formation of these digital competencies). That is:

- Research activities: collection and analysis of information using UAVs, artificial intelligence technologies in genetics, breeding, seed production and biotechnology of crops

in order to create highly productive varieties and hybrids; experiments using digital twins of biological systems; generalization and analysis of the results of scientific experiments using packages of applied analytical programs; mathematical modeling of processes based on standard software packages; preparation of data for reporting, reviews and scientific publications based on the use of modern text editors;

- Organization and management: making managerial decisions on the implementation of technologies for the cultivation of new varieties or hybrids of agricultural crops in various economic and weather conditions based on the use of decision support system (DSS); marketing research in agricultural markets using information technology; control over compliance with technological and labor discipline, including with the use of MES- and geo-information systems;

- Production and technological activities: substantiation of the choice of crop varieties for the specific conditions of the region and the level of intensification of agriculture, and the preparation of seeds for sowing using computer databases; compiling soil-cultivating, sowing and harvesting units and monitoring the patterns of their movement through the fields with GPS navigation techniques; organization of a crop rotation system, its positioning on the territory of land use of an agricultural company and the MES-based cutting of fields.

## 4. Discussion

Nowadays, agricultural companies in most countries have been choosing "industry 4.0" ("agriculture 4.0"). Industry 4.0 is an industrial revolution that involves a new approach to production based on the mass introduction of information technologies into the economy, large-scale automation of business processes, and the spread of artificial intelligence. Agricultural companies are adopting industry 5.0. This is a cyber-social system that allows combining human and machine intelligence to create a collective super intelligence, being a source of harmonious, technological development of human civilization [19].

Numerous research papers note various aspects of the latest digital technologies used today in agriculture and the need to restructure the training of bachelors and masters in order to meet new realities of modern production.

We have previously reviewed up-to-date digital technologies in crop production [20]. Currently, the main trend in the development of agriculture is the use of precision farming technologies based on digital equipment and technology [21].

Today, technologies such as satellite navigation, unmanned and aerial vehicles (UAVs) equipped with various sensors, IoT platforms and big data, and others have already become ingrained in routine agricultural production. Precision farming technologies are widely used in agriculture in all regions of the Russian Federation. This is facilitated, in particular, by the development of unmanned aircraft [22–25]. Machine vision and artificial intelligence technologies [26–30], big data and machine vision are used to predict crop yields [31–34]. The management of agricultural production is developing along with cyber-physical systems using all kinds of sensors [35,36]. This necessitates the restructuring of business processes in the industry and, accordingly, production management, which, in turn, puts forward new requirements for the qualifications of specialists and for the training of graduates of agricultural universities.

The authors of [37] speak about the need to organize a modular approach to the formation of bachelors' competencies and determine the level of formation of each of them, highlighting advanced, basic and threshold indicators at the levels of "knowledge-ability-skills-action", which will improve the effectiveness of the educational process.

The authors of [38] rightly note that universities should constantly be in search of new market needs for the digital competencies of specialists. In our opinion, this should be carried out about once every six months. It is this period that is relevant today for the emergence and promotion of new digital technologies in agricultural production.

The authors write that change in the technological order, and the transition of agriculture to a digital economy, today require universities to take adequate action, to form

competencies that have only recently been included in federal educational standards, which today requires a radical reworking of the main educational programs for in-depth study of digital technologies.

The authors have mentioned above, that as the volumes of information have increased, the ability to find the main points in a large amount of information should be added to the new students' competencies. However, in our opinion, today it can be re-placed by the competency of students using various computer services based on big data processing, involving processing of information (including verbal plans) based on mathematical algorithms in order to find the main trends, patterns, etc. From our point of view, in order to bring students' competencies as close as possible to the current market requirements, training should apply a practice-oriented approach, which involves the implementation of such steps as: opening basic departments (on the basis of leading agricultural IT companies), and including representatives of the IT business in the boards of directors of university trustees. The following measures should be used, such as the involvement of employers' representatives in the IT sector in teaching subjects of the professional block, as well as the more active use of such forms of practical training with students as on-site classes.

This, in turn, necessitates the appropriate advanced training of teachers, which, in our opinion, should upgrade the existing qualification of an advanced user.

The conducted research is consistent with the results of individual scientists [39–41].

## 5. Conclusions

Thus, one of the obstacles to the introduction of digital technologies in agricultural production is the lack of specialists with digital competencies. The transformation of the educational process in agricultural universities in accordance with the modern realities of agricultural production requires identification and classification of these production requirements based on a scientific approach, substantiating the competency model of the graduate [42]. According to the research, representatives of the real sector of the economy consider such indicators as the use of digital technologies in education (94%) and the quality of student training (84%) as most important for graduates. A survey of potential employers of leading agricultural companies showed the most sought-after competencies of graduates (most-to-less important): know the main trends in the development of modern digital technologies in crop production; use information systems to develop production plans and control the execution of work, as well as to prepare reports; know the best practices in the use of digital technologies in agronomy (in Russia and abroad); possess skills in working with information systems for managing agricultural production (ERP-systems); and, use mobile applications for visual control of the state of crops.

In our opinion, in order to bring the competencies of students as close as possible to the current market requirements, training should apply a practice-oriented approach, which involves the implementation of such steps as: opening basic departments (on the basis of leading agricultural IT companies), and including representatives of the IT business in the boards of directors of university trustees. In our opinion, measures should be used such as the involvement of representatives of employers in the IT sector in teaching the disciplines of the professional block, as well as more active use of such forms of practical training with students as on-site classes.

At the same time, universities have certain problems with the implementation of this task, related to the need to improve the digital skills of teachers and fill the materials and resources of agricultural universities with new digital technology.

Since more and more new technologies are being introduced into agricultural production, new competencies of agricultural specialists are vital. Therefore, further scientific research should be carried out every year in order to identify the current requirements of production and the corresponding improvement of the curricula for the training of students in agricultural universities.

**Author Contributions:** Data curation, E.K.; Formal analysis, E.K. and M.N.S.; Investigation, E.K.; Supervision, E.K. and M.N.S.; Writing—original draft, E.K.; Writing—review & editing, A.G. and A.S. All authors have read and agreed to the published version of the manuscript.

**Funding:** This research was funded by the Ministry of Science and Higher Education of the Russian Federation in accordance with agreement No. 075-15-2020-905 date 16 November 2020 on providing a grant in the form of subsidies from the Federal budget of the Russian Federation. The grant was provided for state support for the establishing and development of a world-class Scientific Center "Agrotechnologies for the Future".

**Institutional Review Board Statement:** Not applicable.

**Informed Consent Statement:** Informed consent was obtained from all subjects involved in the study.

**Data Availability Statement:** Full data of this study can be provided upon request.

**Acknowledgments:** The research was supported by the Ministry of Science and Higher Education of the Russian Federation in accordance with agreement No. 075-15-2020-905 date 16 November 2020 on providing a grant in the form of subsidies from the Federal budget of the Russian Federation. The grant was provided for state support for the establishing and development of a world-class Scientific Center "Agrotechnologies for the Future".

**Conflicts of Interest:** The authors declare no conflict of interest.

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
