# Peer review of "Requirements of Modern Russian Agricultural Production for Digital Competencies of an Agricultural Specialist"

_education, doi:10.3390/educsci13020203_

Round 1

Reviewer 1 Report

Title: consider amending the title that fits better the content

The Abstract should report the aim of the study presented in this paper, and be specific on the methodology used, on the research area and the period of the study. A conclusion Is missing as well as further needed research in this field

Keywords: Russia is missing

11 – you meant at least (minimum) once a year?

14-20: consider the use of i; ii; iii; iv

Introduction: the introduction needs to be expanded and be built more on the existing knowledge about the topic in the study area/country a and in worldwide and based on a larger literature review. The objective of the research should be written more clearly, adding research questions

Lines

28: references should appear in order (start with reference 1)

32: reference is missing here

35: can you specify which digital skills? 40% of the rural population?

Table 1: the explanation why these are the products that have been chosen and why is missing. A Reference is missing.

51 – Table 2: reference is missing

48: the objective of the study should specify the country/area of research

77 – the goal should be restated

85: please explain what you mean by scientific comparison

97: was the quantitative data also compared with qualitative data referred to the needed competence in digital crop production technologies? 

99 specify the type of companies and enterprises. This should go to the materials and methods

110-117: it should be included in the material and methods

Table 4: was there an open question to know if the participants wanted to share a different view? Explain the score system

Fig 1. What is the n=? Explain the score system. Are these only digital competences?

156-168: are these learning outcomes all the sam level or can you prioritise some according to your results?

175: which ones?

Further results could be presented in further tables/graphs.

Discussion

220 : in the discussion more comparison between your study results and the literature should be done. Explain what is the Industry 4.0 and Industry 5.0

225-227: references should be cited here

257: is your opinion here based on your research results? Please explain which ones and how

257-269: this should go in the conclusion

Conclusion

275:  references needed

288: explain what it is the way forward, how the situation can improve, need for more research, etc.

Author Response

Dear Reviewer
We took into account your recommendations and made the following changes to the manuscript:
1. Changed the title of the article and made changes to the text.
2. The annotation of the manuscript has been specified;
3. Checked and confirmed citation references for relevance to the text of the manuscript.
4. The hypotheses of the study correspond to the plan, are confirmed by the result of the study.
5. Conclusions corrected and balanced.

With respect and gratitude for your time, Authors of the manuscript

Reviewer 2 Report

The manuscript seems original, the authors' own work, evidenced by the rather small percentage of 15% similarity in anti-plagiarism software (see the attached pdf file).

Please allow me to make some suggestions/recommendations to the authors to improve the manuscript for publication:

I recommend the authors to follow the template of this journal found on the Instructions for Authors page

https://www.mdpi.com/journal/education/instructions

There it is clearly specified that: "References should be numbered in order of appearance...". Unfortunately, in this manuscript the references start with [3], then [1], then [5,13,17,20,26], etc.

I recommend the authors to try not to break the tables on different pages, especially the small ones (for example table 1).

I suggest the authors to decide which separator symbol for decimals to use, the comma ","or the period "." (I for one recommend the period "." for the English format). The authors use in the text the period "." but in Figure 1 they use the comma ",".

It seems to me that the list of bibliographic references is rather thin. I advise the authors to add also other bibliographic references that could emphasize some more the good knowledge of the studied field. Both in the Introduction but also in the section of 2.Materials and Methods which seems quite short for the analyzed phenomenon related to current digital competencies of a modern agricultural specialist.

Also, I would have liked to see how the authors emphasize their contribution a bit more, in comparison, vis-à-vis the other studies and articles in the field, even in the Discussions section.

I would have liked to read in this manuscript also about other technologies with possible implications in agricultural production, something more up-to-date compared to ERP systems (which I've been teaching and researching in the last 20 years). These can be combined, for example, with RPA (Robotic Process Automation), with AI (Artificial Intelligence), blockchain, Big Data, etc. I understand that since 15th of November we are already officially 8 billion inhabitants on the planet, and an increase in production and productivity in agriculture would be desirable.

I think it would be interesting to read in this manuscript the possible limitations of the study carried out by the authors, but also possibilities for improvement, to continue the research, something more about the future plans of the authors.

Author Response

Dear Reviewer
We took into account your recommendations and made the following changes to the manuscript:
1. The numbering of links has become correct.
2. Breakdown of tables is excluded.
3. Symbols denoting fractions of units are brought into unity.
4. The bibliographic list has been expanded.
5. Descriptions of similar technologies used in agricultural production are given.
6. Possible limitations of the studies are indicated.

With respect and gratitude for your time, Authors of the manuscript

Round 2

Reviewer 1 Report

Dear Authors,

I reckon your manuscript has improved in clarity and depth by addressing the suggested comments. Please make sure that the abstract has the right lenght. Some of the details currently in the abstract can be included in the Introduction section. Some English language revision may be needed. 

Author Response

Dear Reviewer. Thanks a lot for your recommendations. We have taken them into account in the revised version of the article: the abstract has been shortened (196 words), the English language has been proofread. Sincerely, the team of authors.

Reviewer 2 Report

From what I see in the new version of the manuscript, the authors have improved it.
However, one of the recommendations I made last time was not taken into account. I quote myself:
I suggest the authors to decide which separator symbol for decimals to use, the comma ","or the period "." (I for one recommend the period "." for the English format). The authors use in the text the period "." but in Figure 1 they use the comma ",".

7,5 from Figure 1 should be 7.5
8,5 should be 8.5
9,5 should be 9.5

so that all the decimal numbers from the manuscripts should be in a unitary decimal format.

See the attached file with red colors.

Author Response

We did not take into account the correction in the drawing because we missed it from attention. We apologize and thank you for your attentiveness and the extra time to review the article.